# Modulation of Plasmatic Matrix Metalloprotease 9: A Promising New Tool for Understanding the Variable Clinical Responses of Patients with Cystic Fibrosis to Cystic Fibrosis Transmembrane Conductance Regulator Modulators

**DOI:** 10.3390/ijms241713384

**Published:** 2023-08-29

**Authors:** Michela Capraro, Marco Pedrazzi, Roberta De Tullio, Marcello Manfredi, Federico Cresta, Carlo Castellani, Monica Averna

**Affiliations:** 1Department of Experimental Medicine (DIMES), University of Genova, 16126 Genova, Italy; capraro36@gmail.com (M.C.); marco.pedrazzi@unige.it (M.P.); detullio@unige.it (R.D.T.); 2Department of Translational Medicine, University of Piemonte Orientale, 28100 Novara, Italy; marcello.manfredi@uniupo.it; 3Cystic Fibrosis Center Genova, IRCCS Istituto Giannina Gaslini, 16147 Genova, Italy; federicocresta@gaslini.org (F.C.); carlocastellani@gaslini.org (C.C.)

**Keywords:** Trikafta^®^, NF-kB, ERK1/2, personalized medicine, circulating mononuclear cells (CMCs), cystic fibrosis, biomarker

## Abstract

Background: The most recent modulator combination, elexacaftor/tezacaftor/ivacaftor (Trikafta^®^), has been shown to improve clinical outcomes in most patients with cystic fibrosis (PwCF). Unfortunately, the clinical benefits are sometimes variable; thus, improving our knowledge of the possible causes of this variability can help reduce it. Methods: Circulating mononuclear cells (CMCs) and plasma were collected from 16 PwCF (including those on Trikafta^®^ therapy) and 4 non-CF subjects. Cystic fibrosis transmembrane conductance regulator (CFTR) activity and matrix metalloprotease 9 (MMP9) expression were monitored before and after therapy, together with some clinical parameters. The relationship between MMP9 expression and the modulation of the extracellular-regulated 1/2 (ERK1/2) and nuclear factor-kB (NF-kB) pathways was also analyzed. Results: MMP9, markedly expressed in the CMCs and plasma of all the patients included in the study, was downregulated in the clinically responsive PwCF. In the non-responder, the MMP9 levels remained high. The modulation of MMP9 following treatment with Trikafta^®^ may be controlled by the NF-kB pathway. Conclusions: These data strongly suggest that MMP9 downregulation is a potential biomarker of therapy efficacy and that it could be useful in understanding the molecular events underlying the variable clinical responses of patients to Trikafta^®^. This knowledge could be helpful for future studies of personalized medicine and thereby ensure improvements in individual responses to therapies.

## 1. Introduction

Cystic fibrosis (CF) is an autosomal recessive disease caused by mutations in the cystic fibrosis transmembrane conductance regulator (CFTR) gene [1,2,3]. The respiratory and gastrointestinal tracts [4,5,6] are particularly affected. However, since CF is characterized by the persistence of inflammation, the impairment of cellular immunity, represented primarily by monocytes, macrophages, and dendritic cells (DCs), is often discussed [7,8,9,10,11]. Different human studies have demonstrated enhanced inflammatory cytokine secretion by monocyte-derived macrophages in CF [12,13]. Furthermore, other studies have reported a direct link between functional CFTR and the anti-inflammatory M2 polarization of macrophages [14]. Others have reported that CF monocytes, due to CFTR dysfunction, overexpress matrix metalloprotease 9 (MMP9) [15,16], which, along with other MMPs, is expressed in inflammatory cells and regulates inflammation in various tissues and diseases [17,18,19,20]. Moreover, MMP9, a potent extracellular matrix (ECM)-degrading enzyme, is considered one of the main effector proteases of tissue remodeling in CF [21,22,23,24]; it not only produces ECM cleavage products, but also processes cytokines which have been implicated in leukocyte infiltration [25]. As a result, the airways of PwCF patients are characterized by protease activation, which contributes to subsequent airway remodeling and disease pathogenesis. To determine whether and how the immune cells are modified in response to CFTR modulator therapy, we recently analyzed, using a proteomic approach, freshly isolated circulating mononuclear cells (CMCs) following both ex vivo and in vivo ivacaftor treatment [26]. The results we obtained demonstrated that the therapy downregulated intracellular MMP9, and that this protein modulation correlated with the improvements in the forced expiratory volume in 1 s (FEV1) and sweat chloride values of the patients treated with the therapy. In line with these results, it has been reported that MMP9 expression has been reduced in patients who positively responded to lumacaftor/ivacaftor therapy [27]. These data were obtained via an RNA-Seq evaluation of whole blood gene expression changes in response to the therapy.

It has been reported that elevated levels of MMP9 are associated with lung function decline in PwCF [21,22,23,24], and many CF airway macrophages are derived from circulating monocytes [24,28]. Thus, after ivacaftor therapy, the downregulation of MMP9 in CMCs could represent one of the possible positive effects of these CFTR modulators in decreasing disease progression, and it could therefore represent a potential biomarker of the efficacy of the CF therapy.

Finding non-invasive biomarkers of an individual patient’s response is crucial in CF. Indeed, although care for PwCF has been revolutionized by the development of CFTR modulators [29,30], clinical responses are often variable [31,32,33].

In this study, we have measured MMP9 levels in both the CMCs and the plasma of responders and non-responders to Trikafta^®^, the triple CFTR modulator therapy. Thus, we intend to obtain more information about the modulation of MMP9 and confirm whether the downregulation of MMP9 in CMCs could be a valuable biomarker of CF modulator therapy efficacy. Moreover, we also investigated the modulation of MMP9 expression, since the signaling mechanism for the increase in MMP9 in CF is poorly understood. In this context, we previously reported that in CF CMCs, the activation of the protein kinase C/ERK1/2 pathway, which is promoted by the altered intracellular calcium homeostasis, induces MMP9 expression [16]. Moreover, other authors have reported that, in different CF cellular models, the absence or reduction CFTR activity promotes the activation of cross-talk between the ERK pathway and nuclear factor-kB (NF-kB) [34,35]. Thus, we investigated the possible involvement of the phospho-(p-)ERK1/2/NF-kB signaling pathway in the increase in MMP9 expression in the CMCs of PwCF and the modulation of protease expression after Trikafta^®^ therapy.

## 2. Results

### 2.1. MMP9 Expression in CMCs of PwCF during Trikafta^®^ Therapy

Although CFTR modulators have revolutionized the care of patients with CF, clinical responses to these molecules are variable and sometimes absent. Unfortunately, also Trikafta^®^, which is approved for the treatment of most PwCF, sometimes fails as well. Table 1 shows the clinical parameters of five PwCF who were monitored before and after five and ten months of Trikafta^®^ therapy. Four of the five patients considered for this study showed improvements in some of the clinical parameters, though these remained unchanged in one patient. The levels of CFTR activity in the CMCs of the each PwCF were measured at the same time points during the therapy (see Table 1).

As Table 1 shows, the NaI exchange values agree with the clinical data, and the only patient who did not improved clinically was the one who did not show any increase in CFTR activity after the therapy. Thus, this patient was considered a non-responder (CF85), while the other four were considered responders (CF108, CF161, CF76, and CF157). To understand the reason for this variability in drug response, we decided to monitor the expression of MMP9 in the CMCs of all five patients who underwent the therapy. The rationale for this was based on the results we obtained in previous research, in which we showed that MMP9 is upregulated in the leukocytes of F508del^+/+^ patients [16].

Thus, we first monitored the MMP9 in the CMCs before the Trikafta^®^ therapy, not only in the 5 PwCF just considered, but also in 11 other patients who had undergone the therapy for less time and on whom we have not yet collected sufficient information.

As is shown in Figure 1, MMP9 was expressed in the CMCs of all the PwCF, whereas no MMP9 was detectable in the CMCs from 4 non-CF subjects. These results agree with our previously collected data that were used to analyze MMP9 expression in the CMCs of PwCF homozygous for F508del-CFTR [16]. Since we had also reported that the downregulation of MMP9 could be a potential leukocyte biomarker related to ivacaftor efficacy [26], we then measured the MMP9 levels in the CMCs of the non-responders and responders before and after the indicated times during the Trikafta^®^ therapy.

As Figure 2A,B show, intracellular MMP9 expression in the responders showed a downregulation trend after both therapy periods (POST I and POST II). This did not occur in the non-responder. Although these are only preliminary results, they do not indicate that there is any time course for treatments acting on MMP9 expression. Indeed, there was no significant difference between the POST I and POST II results in any of the samples considered (Figure 2B). However, the intracellular MMP9 expression values obtained from all the responders at POST I and POST II were significantly downregulated after the Trikafta^®^ treatment, but those of non-responder were not.

These results confirm the potential of MMP9 downregulation in the leukocytes as a biomarker of CFTR modulator efficacy, and suggest that it can be used to evaluate Trikafta^®^ therapy.

### 2.2. Identification of Intracellular Pathways Related to MMP9 Expression in the CMCs of the Non-Responder and the Responders

It is known that MMP9 is the most abundant MMP found in bronchopulmonary secretions derived from PwCF [20], that neutrophils constitutively express MMP9, and that other cellular types, such as macrophages and epithelial cells, can produce MMP9 [18]. However, the signaling mechanism for the increase in MMP9 in CF has yet to be well elucidated. Since it has been reported that MMP9 expression can be promoted by the ERK1/2 and NF-kB signaling pathways [34,35], we monitored the possible involvement of these signaling mechanisms in the increase in MMP9 expression in the CMCs of the PwCF as well as the modulation of the protease expression after the Trikafta^®^ therapy.

As Figure 3 shows, the p-ERK1/2 levels in the CMCs of both the non-responder and the responders before the Trikafta^®^ therapy were higher than in those in the CMCs collected after the therapy (POST I and POST II). Even though no significant difference was observed between the dephosphorylation of the two kinases in the non-responder and in the responders (Figure 3B), all the PwCF showed a significant decrease in p-ERK1/2 levels (Figure 3C). This result indicates that Trikafta^®^ therapy could lead to the dephosphorylation of p-ERK1/2, despite the efficacy of the therapy. However, since p-ERK1/2 induces MMP9 expression in leukocytes, as we have previously reported, the dephosphorylation of p-ERK1/2 is in line with the MMP9 modulation data we obtained from the CMCs of the responders, though not with those obtained from the CMCs of the non-responder. To clarify this discrepancy, we studied the downstream events related to the NF-kB pathway, since it has been reported [34] that this nuclear transcriptional factor promotes MMP9 expression. Thus, we monitored the modulation of IkBα because its phosphorylation and degradation allow NF-kB to translocate into the nucleus and bind to the specific MMP9 promoter sequence. As is shown in Figure 3A,D, the amount of IkBα in the CMCs of the responders significantly increased after the Trikafta^®^ therapy, while it decreased in the CMCs of the non-responder. Thus, the modulation of IkBα parallels the modulation of MMP9 expression, indicating that the therapies which target the recovery of CFTR activity promote the downregulation of MMP9 through the NF-kB pathway.

These results, together with those previously obtained [26], indicate that the expression of MMP9 is related to the response of the PwCF to CFTR modulators, and that by elucidating the NF-kB signaling pathway, which is related to MMP9 expression, it may be possible to understand the molecular events underlying the variability in the clinical responses of PwCF to their CF therapies.

### 2.3. Levels of MMP9 in the Plasma Samples of the Non-Responder and the Responders

In order to verify whether MMP9 modulation could be a promising biomarker of the efficacy CFTR modulators, we also measured the MMP9 activity before Trikafta^®^ therapy in plasma isolated during the collection of the CMC samples from the 16 PwCF whose intracellular MMP9 expression is shown in Figure 1.

As is shown in Figure 4A,B, before therapy, the plasma samples of all the patients expressed MMP9 activity significantly higher than that observed in the four non-CF subjects. These data agree with those obtained from the analyses carried out on the CMCs and reported in Figure 1. Moreover, we measured the MMP9 activity in the plasma samples of the non-responder and the responders both before and after the therapy.

As Figure 5 shows, the plasmatic MMP9 activity of the non-responder did not change after the therapy. However, that of responders significantly decreased (by about 50%) as it occurred at the intracellular level (see Figure 2B and Figure 5B). Thus, all these results showed MMP9 modulation similar to that presented in Figure 2, suggesting that the MMP9 measured in plasma likely corresponded to that secreted by CMCs.

Finally, to further support the notion that plasmatic MMP9 modulation could be a potential biomarker of CFTR modulator efficacy, we also preliminarily evaluated the changes in plasmatic MMP9 activity in 8 of the 11 PwCF who had started Trikafta^®^ therapy, but whose plasmatic samples and clinical outcomes were already available (see Table 2).

As Figure 6A,B show, seven PwCF, clinically identified as responders, showed, to different extents, the downregulation of plasmatic MMP9 activity at the first follow-up. In contrast, this was not the case for CF73, who was the only one clinically identified as a non-responder. These results, though they are still preliminary, and those previously obtained, further strengthen the hypothesis that plasmatic MMP9 can serve as a promising biomarker of the efficacy of CFTR modulators.

Figure 7 shows a schematic drawing that depicts the effect of Trikafta^®^ therapy on the NF-kB pathway involved in MMP9 upregulation detected in the CMCs of the PwCF. Briefly, the increase in [Ca^2+^]_i_ that was observed in the CF cells as a consequence of reduced CFTR activity [16], as well as pro-inflammatory signals, can lead to ERK 1/2 activation, which in turn promotes the release of NF-kB from its complex with IkBα through the phosphorylation and subsequent ubiquitination/degradation of the inhibitory protein. Our results suggest that Trikafta^®^ therapy acts at this pathway stage. Once they are no longer complexed with IkBα, NF-kB subunits can translocate to the nucleus, promoting MMP9 mRNA transcription and, thus, MMP9 protein upregulation. Consequently, MMP9 can be detected in the extracellular environment and in the plasma of PwCF. Finally, the effect of the therapy on this molecular pathway led to significant MMP9 downregulation in those patients that were identified as responders. Conversely, in the non-responder’s CMCs, although the dephosphorylation of ERK1/2 still occurred, the phosphorylation/degradation of IkBa persisted, preventing the downregulation of MMP9 expression.

## 3. Discussion

In this study, we found that MMP9, which was upregulated in the CMCs and plasma of five PwCF before Trikafta^®^ therapy, was downregulated at different times following the therapy in the biological samples of four clinically positive responders. By contrast, in the non-responder patient, this protease remained unaffected. The downregulation of MMP9, detected after the therapy, paralleled the recovery of CFTR activity assayed in the same CMC samples, and with the amelioration of the clinical outcomes. We also showed that this modulation in intracellular MMP9 expression, and the subsequent secretion of MMP9 in the plasma, could be controlled by NF-kB activation/inhibition through the degradation/synthesis of its inhibitor, IkBα. In particular, the CMCs of the responder patients showed increased IkBα levels after the therapy, when both intracellular and plasmatic MMP9 levels decreased. Conversely, the intracellular level of IkBα in the non-responder decreased even when their MMP9 remained over-expressed after the therapy. Thus, MMP9 expression could be promoted by the translocation to the nucleus of NF-kB due to the phosphorylation/polyubiquitination/degradation by the proteasome of its cytosolic inhibitor, IkBα, as was reported in [36]. Several reports have shown a relationship between increased NF-kB activation [37,38,39] and CFTR dysfunction in various CF cell lines which can intrinsically cause the expression of pro-inflammatory mediators, but less attention has been paid to studying the relationship of this pathway with MMP9 expression in CF CMCs. In this context, we also found that ERK1/2 phosphorylation was significantly higher in the CMCs of both responders and the non-responder before the therapy. This result is consistent with the possible involvement of p-ERK1/2 in the phosphorylation process of IkBα before its degradation. We previously reported a relationship between the p-ERK1/2 levels and MMP9 expression in CF [16], in agreement with other studies [40]. However, we now suggest that the downregulation of MMP9 expression in the CMCs and plasma of responders to Trikafta^®^ therapy seems to be due to a decrease in p-ERK1/2 which is sufficient to promote a consequent increase in IkBα and inhibit NF-kB translocation into the nucleus. In contrast, in the CMCs of the non-responder patient, the decrease in p-ERK1/2 detected after the therapy failed to increase the level of IkBα, and, thus, to downregulate MMP9 expression, suggesting the involvement of other kinases in the IkBα phosphorylation process, as has already been reported in the literature [35,36]. Further studies, possibly performed using a proteomic approach, are recommended to support this hypothesis and better clarify the discrepancy in the p-ERK1/2/NF-kB signaling pathway and MMP9 expression observed in the responder and non-responder PwCF.

To strengthen the significance of the data we obtained, we extended the analysis of MMP9 in CMCs and plasma to 11 other patients who had not yet undergone Trikafta^®^ therapy. Furthermore, among these patients, we selected eight who, in the meantime, had started Trikafta^®^ therapy, and measured their plasma levels of MMP9 after the initial phases of the therapy. The results we obtained, along with data concerning the clinical conditions of the patients, show that the CMC and plasma samples had similar MMP9 modulation, and therefore suggest that the plasmatic downregulation of MMP9 can be a biomarker of positive responses to Trikafta^®^ therapy. These data agree with our previous results [11,26], obtained from CMCs following both ex vivo and in vivo treatment with another CFTR modulator (ivacaftor). Notably, using a quantitative proteomic and bioinformatic approach, we identified several metalloprotease proteins usually involved in cystic fibrosis [21], such as matrix metalloproteinase 8 (MMP8), MMP9, and matrix metalloproteinase 16 (MMP16). However, only MMP9 was downregulated after the treatment.

MMP9, which is expressed by several immune cells, such as monocytes/macrophages and neutrophils [41,42], has been detected in association with CF progression [18,20,28]. In addition, MMP9 levels are higher during CF progression, which is characterized by an acute pulmonary exacerbation associated with increased inflammation due to the recruitment of many monocytes to the site of the inflammation and damage [28,43]. Overall, MMP9 levels are positively correlated with the degradation of basement membrane collagen and a decline in lung function in PwCF [44,45]. Thus, the downregulation of MMP9 could represent one of the possible positive effects of CFTR modulators in decreasing disease progression. In support of this hypothesis, it has been reported that, based on an RNA-Seq evaluation of whole blood gene expression changes in response to lumacaftor/ivacaftor, MMP9 expression was reduced in those patients that positively responded to the therapy [27]. Moreover, during our plasma sample analysis, we sometimes observed a double band of plasmatic MMP9 activity because MMP9 is secreted in the extracellular space as an inactive pro-enzyme named pro-MMP9 (92 kDa). However, in the extracellular space, other proteinases, such as MMP3 and MMP2, cleave the inactive form pro-MMP9 into the active form 84 kDa [46,47]. We can see the MMP9 double band only in those plasma samples which show more significant quantities of MMP9 than the others. However, an analysis of the active band of MMP9 in the plasma and its relationship with other metalloproteases will have to be the subject of a future study.

Our results, obtained by monitoring MMP9 levels directly in the CMCs and plasma of PwCF undergoing Trikafta^®^ therapy, provide additional information which is needed to validate MMP9 modulation as a biomarker of CF therapy efficacy. Moreover, we investigated the relationship between the p-ERK1/2/NF-kB signaling pathways and MMP9 expression, highlighting a potential alteration in the modulation of ERK1/2 phosphorylation in the CMCs of the non-responder patient. In order to elucidate the biochemical mechanisms related to the NF-kB signaling pathway and MMP9 expression, we will carry out future studies on more suitable cell models (e.g., monocytes/macrophages) directly isolated from responders and non-responders to the therapy. The data thus obtained will be helpful in better understanding the molecular events underlying the variable clinical responses of PwCF to CFTR modulators. This knowledge, obtained via a simple blood draw, can be helpful for future studies of personalized medicine and ensure improvements in individual responses to therapies. Further analyses involving additional non-responder patients are necessary to strengthen these results.

## 4. Materials and Methods

### 4.1. Materials

RPMI 1640, Lympholyte^®^-H, and pre-stained protein SHARPMASS VI MW marker were purchased from Euroclone SpA (Milan, Italy). Anti-MMP9 antibody, anti-P-ERK 1/2 antibody, anti-ERK1/2 antibody, horseradish peroxidase (HRP)-linked anti-rabbit and anti-mouse secondary antibodies, protease inhibitor cocktail (100×), and phosphatase inhibitor cocktail (100X) were obtained from Cell Signaling Technology (Danvers, MA, USA). Anti-IkBα and anti-β-actin antibodies were obtained from Santa Cruz Biotechnology Inc. (Dallas, TX, USA). Dibutyryl-cAMP, the potentiator ivacaftor (VX770), gelatin, Triton X-100, and BRIJ^®^35 Detergent Calbiochem^®^ were purchased from Sigma-Aldrich (Milan, Italy). ECL Select™ Western Blotting Detection Reagent, ECL Western Blotting Detection Reagent, and Amersham™ Protran^®^ Premium 0.45-µm nitrocellulose were obtained from GE Healthcare (Chicago, IL, USA). Brillant Blue R-250 and Acrylamide/Bis Solution were obtained from Bio-Rad Laboratories Srl (Segrate, MI, Italy).

### 4.2. Ethics Statement

All the participants gave their written informed consent prior to their inclusion in the study, including permission to store the samples and to use them for research. The study protocol conformed to the provisions of the Declaration of Helsinki and those of the G. Gaslini Children’s Hospital, Genoa, Italy. The Ethics Committee of Genoa approved the study under protocol number A-CF2014 460REG2014.

### 4.3. Donor Subjects and Sample Collection

Our analyses were carried out on blood samples obtained from all the participants during their routine clinical examinations at the hospital. Sixteen PwCF undergoing Trikafta^®^ therapy (see Table 1 and Table 2) and four non-CF donors were enrolled in the study. The clinical conditions of the patients were evaluated further via ppFEV1(%), sweat chloride tests, body mass index (BMI), and pulmonary exacerbation (PEx) frequency. All the PwCF were regularly followed at the Cystic Fibrosis Center, G. Gaslini Hospital, Genova, Italy. For every patient and non-CF donor, samples of approximately 8 mL of blood were collected in three 3 mL vacuette^®^ PREMIUM tubes containing 5 mM EDTA.

### 4.4. CMC Isolation and Plasma Collection

The blood samples were diluted with an equal volume of RPMI 1640 and carefully stratified over the Lympholyte^®^-H Cell Separation Media, following manufacturer’s instructions. Briefly, the stratified samples were centrifuged at 800× *g* for 20 min at 22 °C without a break. After centrifugation, the CMCs were collected at the interface between the upper layer, containing the plasma fraction, and the lower layer, containing the Lympholyte^®^-H. To remove the platelets, the CMCs were washed twice with RPMI 1640 and then resuspended in PBS or CFTR buffer for the relevant subsequent analysis.

### 4.5. CFTR Assay

The measurements of CFTR activity were carried out as described in [26] by means of GST-HS-YFP on vehicle- or cAMP/VX770-stimulated PBMCs. Unknown NaI concentrations were extrapolated from the GST-HS-YFP NaI-quenching curve, and CFTR activity was indicated as NaI exchange and expressed as pmol/min/10^3^ cells.

### 4.6. Western Blot Analysis

Freshly isolated CF and non-CF CMCs were lysed at 10^7^/mL via sonication in Laemmli sample buffer and heated for 5 min at 95 °C, and 30 µL aliquots of each sample were subjected to SDS-PAGE, followed by a Western blot. The nitrocellulose membranes were blocked via incubation for 1 h at room temperature with 5% (*w/v*) skim milk powder in PBS containing 0.05% (*v*/*v*) Tween-20. The membranes were incubated successively for 16 h at 4 °C with each of the following primary antibodies: anti-MMP9 (1:1000), anti-β-actin (1:1000), anti-P-ERK 1/2 (1:2000), anti-ERK 1/2 (1:2000), and anti-IkBα (1:1000). The peroxidase-conjugated secondary antibody (1 h at 22 °C) was anti-rabbit or anti-mouse (1:5000). A stripping and re-probing procedure was applied to test the membranes with all the primary antibodies.

Immunoreactive signals were developed using ECL Select™ Western Blotting Detection Reagent, acquired and quantified using a ChemiDoc™ XRS equipped with Quantity One Image Software 4.6.1 (Bio-Rad Laboratories Srl, Segrate, MI, Italy). Alternatively, ECL Western Blotting Detection Reagent was used.

### 4.7. Zimography Analysis

Plasmatic MMP9 activity was assayed via zymography. Briefly, CF and non-CF plasma samples (1 µL) were diluted in a modified Laemmli sample buffer. The samples were subjected to electrophoresis at 4 °C for 1.5 h, without boiling and reduction, through an 8% (*v*/*v*) polyacrylamide gel copolymerized with 1 mg/mL gelatin. The gel was incubated for 1 h at 25 °C in 0.05 M Tris (pH 7.4) containing 2.5% (*v*/*v*) Triton X-100, washed twice with 0.05 M Tris (pH 7.4), and then maintained overnight at 37 °C in 0.05 M Tris (pH 7.4) containing 10 mM CaCl2, 0.15 M NaCl, and 0.05% (*v*/*v*) BRIJ^®^35 Detergent. The gel was fixed and stained for 2 h with a pre-warmed solution containing 45% (*v*/*v*) methanol, 10% (*v*/*v*) acetic acid, and 0.25% Coomassie Blue R-250. The zymograms were de-stained using 30% (*v*/*v*) methanol-10% (*v*/*v*) acetic acid. The relative levels of MMP9 activity were calculated via computer-assisted planimetry, and the intensity of the MMP9-dependent lytic areas was determined using Quantity One Image Software 4.6.1 (Bio-Rad Laboratories Srl, Segrate (MI), Italy).

### 4.8. Statistical Analysis

Where feasible, the data were presented as mean ± SD and analysed for their distribution by means of a Kolmogorov–Smirnov test (test of normality). The significance of the differences was analyzed via non-parametric or parametric tests, as indicated in the relevant figure legend, using the Prism 4.02 software package (GraphPad Software, San Diego, CA, USA), and at least *p* < 0.05 was considered statistically significant.

## Figures and Tables

**Figure 1 ijms-24-13384-f001:**
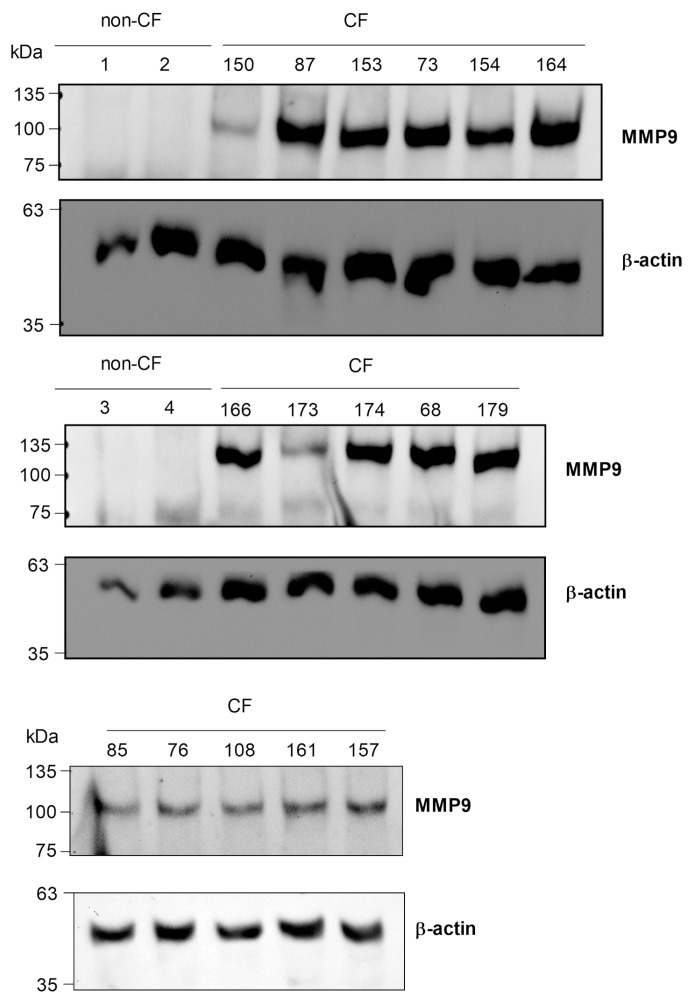
Intracellular MMP9 expression in PwCF and in non-CF donors. MMP9 expression was analyzed in CMCs isolated from both PwCF (*n* = 16) and non-CF donors (*n* = 4) using Western blot analysis. The CMCs were lysed in Laemmli sample buffer and an aliquot of each sample, corresponding to 3 × 10^5^ cells, was subjected to SDS-PAGE (6%), and then a WB analysis for MMP9 and β-actin was performed.

**Figure 2 ijms-24-13384-f002:**
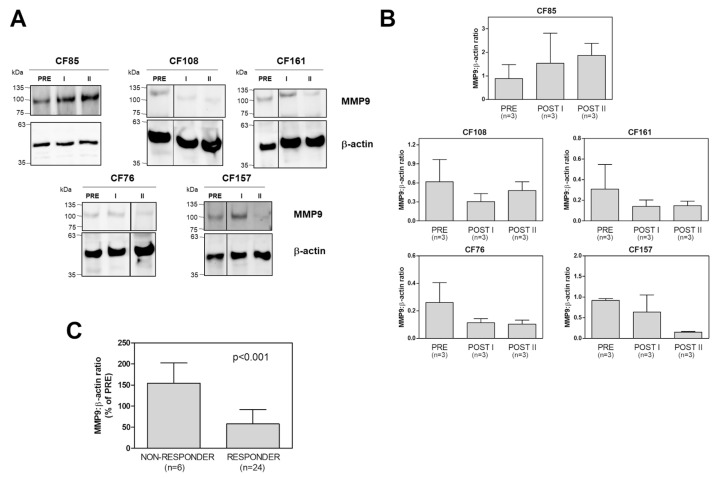
Intracellular MMP9 expression in PwCF following Trikafta^®^ therapy. (**A**) MMP9 expression was monitored in the CMCs isolated from one non-responder patient and from four responder patients, both before (PRE) and after the therapy (I and II), using Western blot analysis. An aliquot of each sample, corresponding to 3 × 10^5^ cells, was subjected to SDS-PAGE (6%), and then a WB analysis for MMP9 and β-actin was performed as a loading control. One representative blot of three is shown. (**B**) For each PwCF, the MMP9 detected in (**A**) was quantified and the MMP9:β-actin ratios were reported as means ± SD (*n* = 3 for each group). (**C**) The immunoreactive signals were quantified and the MMP9:β-actin ratios obtained following the therapy (I + II) were reported as percentages of the ratios before the therapy (PRE) for both the non-responder and the responders. Data are means ± SD from three quantifications for each PwCF (*n* = 6 for the non-responder and *n* = 24 for the responders). *p* < 0.001, according to a Mann–Whitney test.

**Figure 3 ijms-24-13384-f003:**
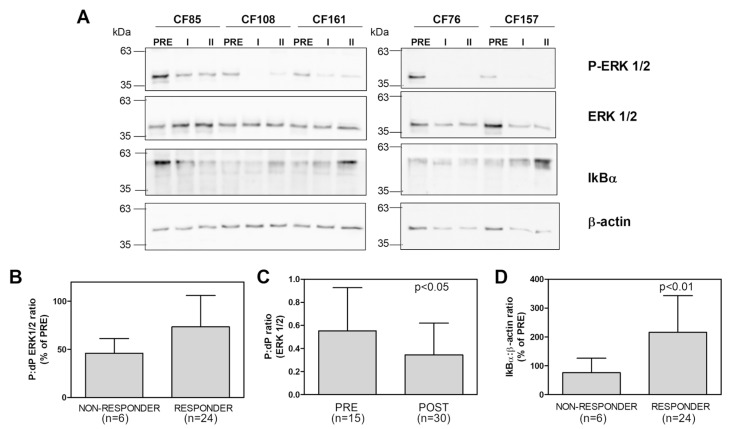
Intracellular signaling pathways related to MMP9 expression in PwCF following Trikafta^®^ therapy. (**A**) ERK1/2 phosphorylation and IkBα modulation were analyzed in the CMCs isolated from one non-responder patient and from four responder patients, both before (PRE) and after the therapy (I and II), using Western blot analysis. An aliquot of each sample, corresponding to 3 × 10^5^ cells, was subjected to SDS-PAGE (10%), and then a WB analysis for the indicated antigens was performed. One representative blot of three is shown. (**B**) The immunoreactive signals were quantified and the ratios of phosphorylated (P) and dephosphorylated (dP) ERK 1/2 forms obtained following the therapy (I + II) were reported as percentages of the ratios before the therapy (PRE) for both the non-responder and the responders. (**C**) The P:dP ERK1/2 ratio was reported and compared with the ratios before (PRE) and after (POST) the therapy. (**D**) The IkBα:β-actin ratios obtained following the therapy (I + II) were reported as percentages of the ratios before the therapy (PRE) for both the non-responder and the responders. Data are means ± SD from three quantifications for each PwCF (the number of the data points is reported for each group). *p* < 0.05 and *p* < 0.01, according to a Mann–Whitney test.

**Figure 4 ijms-24-13384-f004:**
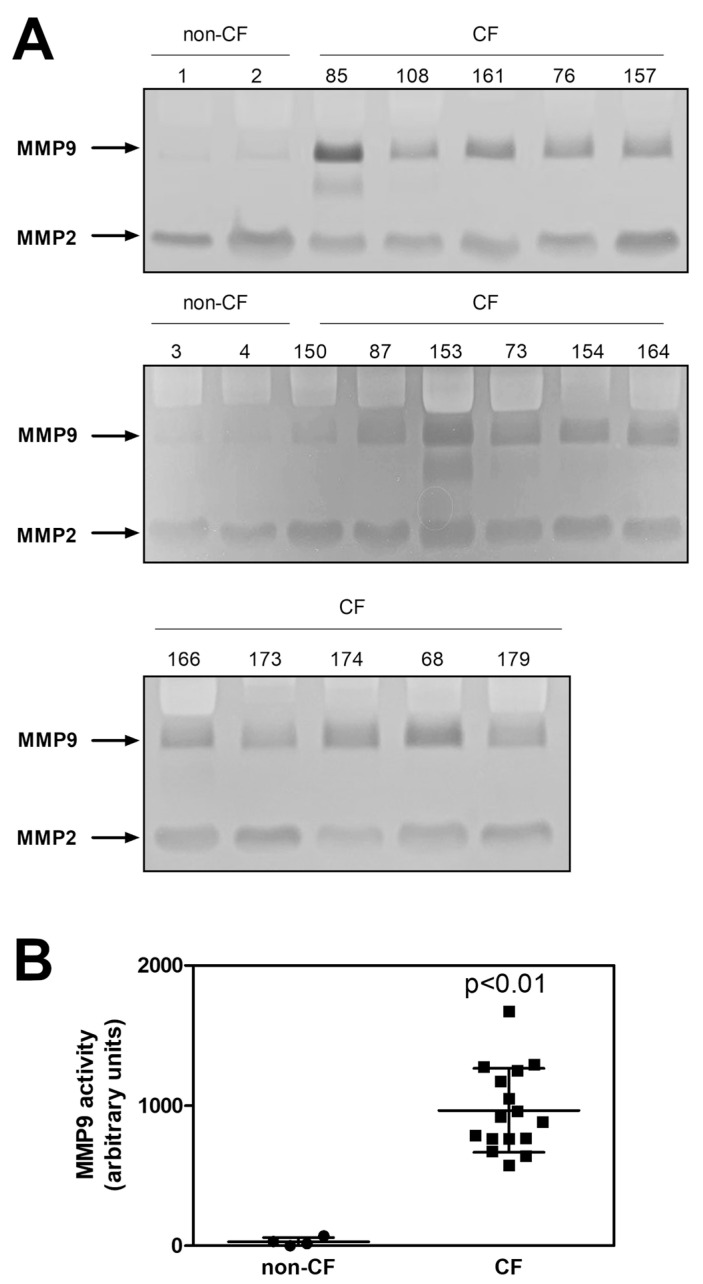
Plasmatic MMP9 activity in PwCF and non-CF donors. (**A**) An aliquot (1 µL) of plasma obtained from the blood samples of the PwCF (*n* = 16) and the non-CF donors (*n* = 4) was subjected to zymography on SDS-PAGE (8%). Arrows indicate the lytic bands corresponding to MMP9 and MMP2 activity. For better visualization, the original images were converted into gray scale and inverted. (**B**) The MMP9-dependent lytic bands were quantified and reported as means ± SD; *p* < 0.01, according to a Mann–Whitney test.

**Figure 5 ijms-24-13384-f005:**
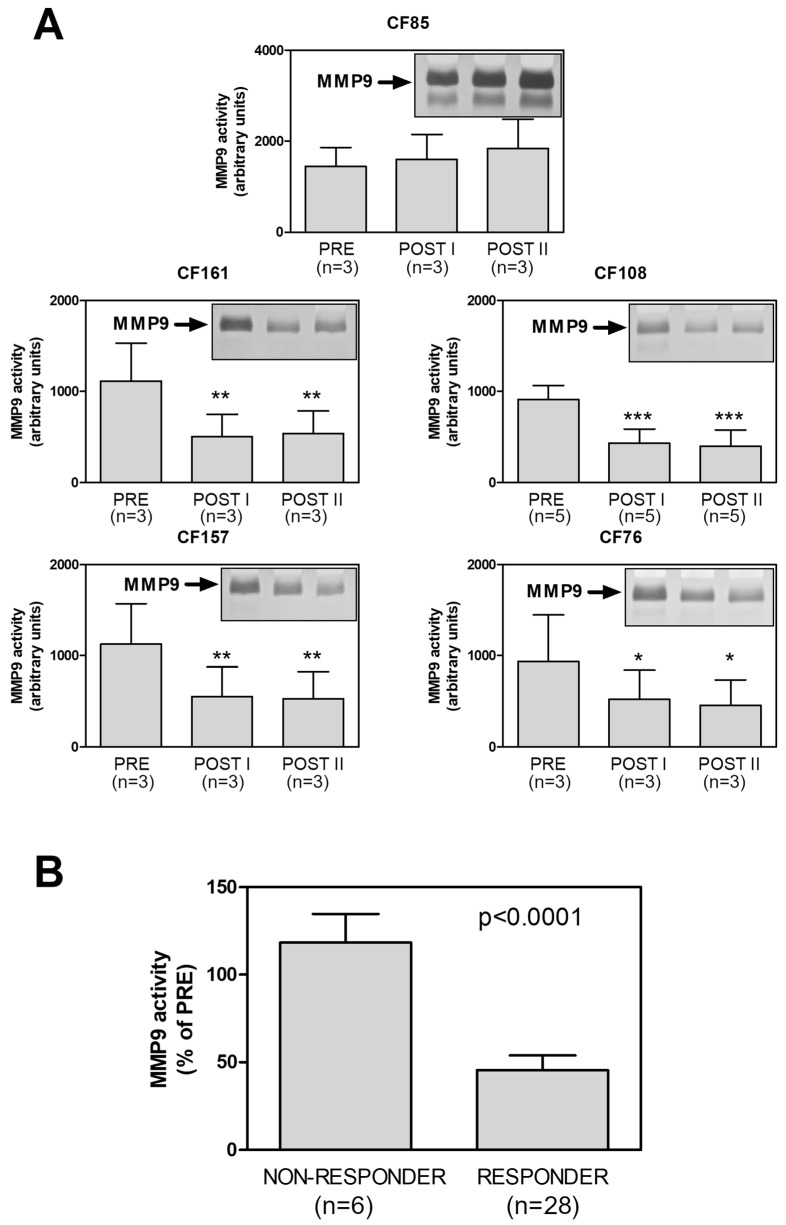
(**A**) Plasmatic MMP9 activity in PwCF following Trikafta^®^ therapy. MMP9 activity was measured in 1 µL plasma samples obtained from the non-responder patient and from the four responder patients, both before (PRE) and after the therapy (POST I and POST II), using zymography. One representative zymogram of three is shown for each patient. The MMP9-dependent lytic bands were quantified and reported as means ± SD from three quantifications for each PwCF (*n* = 3), except for CF 108 (*n* = 5); * *p* < 0.05, ** *p* < 0.01, *** *p* < 0.001, according to an ANOVA followed by a Tukey’s post hoc test. (**B**) MMP9 activity detected following the therapy (POST I + POST II) was reported as a percentage of the metalloprotease activity before the therapy (PRE) for both the non-responder and the responders. Data are means ± SD from three quantifications for each PwCF (*n* = 6 for the non-responder and *n* = 28 for the responders). *p* < 0.001, according to an unpaired *t*-test test.

**Figure 6 ijms-24-13384-f006:**
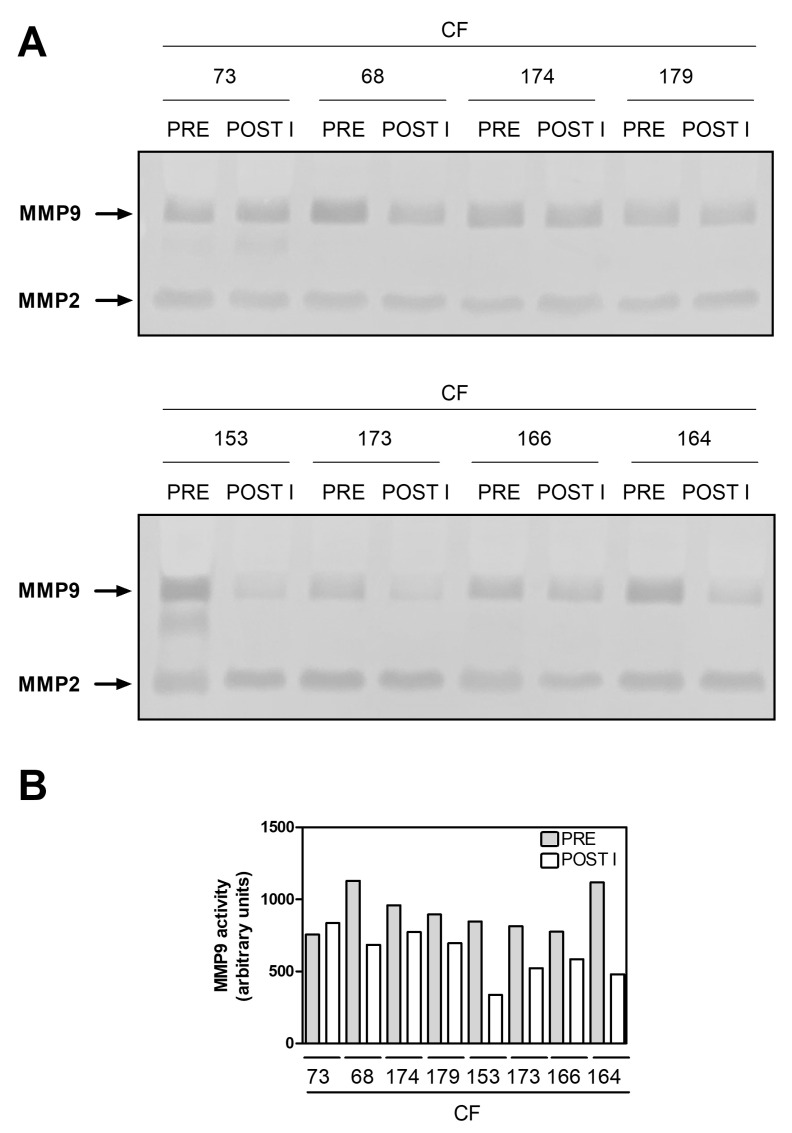
Plasmatic MMP9 activity in other PwCF following Trikafta^®^ therapy. (**A**) MMP9 activity was measured in 1 µL samples of plasma obtained from eight other PwCF, both before (PRE) and after the therapy (POST I), using zymography. (**B**) Quantification of the MMP9-dependent lytic bands shown in (**A**) (*n* = 1).

**Figure 7 ijms-24-13384-f007:**
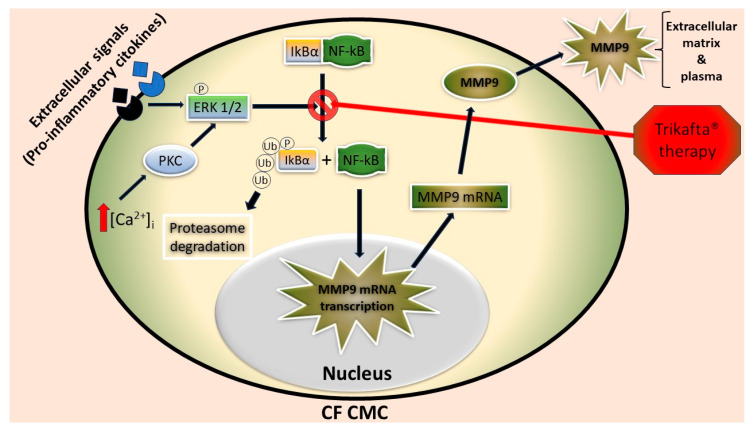
Schematic drawing depicting the possible effect of Trikafta^®^ therapy on the NF-kB pathway that leads to MMP9 upregulation in CF CMCs. [Ca^2+^]_i_: intracellular calcium concentration; P: phosphoric group; Ub: ubiquitin.

**Table 1 ijms-24-13384-t001:** Clinical details of PwCF (both responders and non-responders) to Trikafta^®^ therapy (*n* = 5).

CF Patient Sample Timing	Age	Gender	CFTR Mutation	FEV1 (%)	Sweat Chloride Test *	CFTR Activity ^§^
76-PRE	48	M	F508del/F508del	86	113	ND
76-POST I				83	67	106
76-POST II				78	NA	36
85-PRE	23	F	F508del/E585X	34	104	85
85-POST I				35	96	24
85-POST II				35	97	NA
108-PRE	29	F	F508del/L1065P	69	NA	ND
108-POST I				72	13	189
108-POST II				72	NA	166
157-PRE	14	F	F508del/I1005R	89	NA	13
157-POST I				135	NA	229
157-POST II				122	70	157
161-PRE	16	F	F508del/F508del	67	132	32
161-POST I				113	35	109
161-POST II				113	NA	136

PRE: before the therapy; POST I: five months after the beginning of the therapy; POST II: ten months after the beginning of the therapy; NA: not available; ND: not detectable; * (Cl^−^ mEq/L); ^§^ (I^−^ pmol/min/10^3^ cells).

**Table 2 ijms-24-13384-t002:** Clinical details of other PwCF, including 8 already in the early stages of Trikafta^®^ therapy (*n* = 11).

CF Patient	Age	Gender	CFTR Mutation	FEV1 (%)		Sweat Chloride Test *	
68	34	M	F508del/N1303K	PRE: 62	POST I: 83	PRE: 120	POST I: 94
73	35	M	F508del/G542X	PRE: 32	POST I: 44	PRE: 109	POST I: 109
87	55	F	F508del/R334W	PRE: 77		PRE: 115	
150	31	M	F508del/2184insA	PRE: 83		PRE: 70	
153	17	F	F508del/R553X	PRE:43	POST I: 75	PRE: NA	POST I: 47
154	25	F	F508del/W1282X	PRE: 106		PRE: NA	
164	17	F	F508del/W1282X	PRE: 113	POST I: 152	PRE: 110	POST I: 50
166	22	F	F508del/F508del	PRE: 77	POST I: 99	PRE: 107	POST I: NA
173	14	M	F508del/S13R	PRE: 111	POST I: 101	PRE: NA	POST I: NA
174	14	M	F508del/2183AA>G	PRE: 81	POST I: 89	PRE: NA	POST I: 52
179	31	M	F508del/D110H	PRE: 92	POST I: 107	PRE: NA	POST I: NA

PRE: before the therapy; POST I: five months after the beginning of the therapy; NA: not available. * (Cl^−^ mEq/L).

## Data Availability

Data available on request from the corresponding author.

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
