# Peer review of "Modulation of Plasmatic Matrix Metalloprotease 9: A Promising New Tool for Understanding the Variable Clinical Responses of Patients with Cystic Fibrosis to Cystic Fibrosis Transmembrane Conductance Regulator Modulators"

_ijms, 2023, doi:10.3390/ijms241713384_

Round 1

Reviewer 1 Report

The article by Capraro et al. deals with important issues about an importance of finding non-invasive biomarkers as a clinical response to CFTR modulator therapy in the cystic fibrosis (CF) patients. The importance of biomarkers of an individual response is very crucial in CF thus improving the knowledge of the possible causes related  to this variability. Very interesting is the study of CFTR activity related with the modulation of Matrix metalloprotease 9 (MMP9) expression were monitored before and after therapy together with some clinical parameters. Their work has been adequately designed with a very good description of the methods and their results were presented clearly with conclusions adequately supported. I think that the paper can be considered for publication.

Author Response

The article by Capraro et al. deals with important issues about an importance of finding non-invasive biomarkers as a clinical response to CFTR modulator therapy in the cystic fibrosis (CF) patients. The importance of biomarkers of an individual response is very crucial in CF thus improving the knowledge of the possible causes related  to this variability. Very interesting is the study of CFTR activity related with the modulation of Matrix metalloprotease 9 (MMP9) expression were monitored before and after therapy together with some clinical parameters. Their work has been adequately designed with a very good description of the methods and their results were presented clearly with conclusions adequately supported. I think that the paper can be considered for publication.

Thank you very much for your comment. However, with the suggestions of other reviewers, we modified the first version of the manuscript to improve its scientific content. 

Reviewer 2 Report

The manuscript “Modulation of plasmatic MMP9: a promising new tool to understand variable clinical responses of CF patients to CFTR modulators” describes the evaluation of a potential new non-invasive biomarker for demonstrating clinical effects of tricafta in cystic fibrosis patients with at least one F508-del mutation. The manuscript is well written and the biomarker extensively evaluated. The references are carefully chosen and the subject is of general interest. There are only minor concerns which need to be addressed:

1) The figure legends state: Data are means ± SD from three quantification for each CF patient. Please specify the number of data points per group used for statistical analysis. This is important for understanding the significance of the results.

2) Patient 85 has a double band when staining the membrane for MMP9. Please comment on this in the discussion.

3) Please comment on the fact that the patients are not all homozygous for F508-del. Could their second mutation affect the outcome of the pharmacological treatment?

The English language is overall very good. There are no major issues but double-check "4.7. Zimography analysis".

Author Response

1) The figure legends state: Data are means ± SD from three quantification for each CF patient. Please specify the number of data points per group used for statistical analysis. This is important for understanding the significance of the results.

As suggested, we added the number of data points per group in new Figures 2, 3, and 5.

2) Patient 85 has a double band when staining the membrane for MMP9. Please comment on this in the discussion.

We can observe for patient 85 and also patient 153 a double band of plasmatic MMP9 activity because MMP-9 is secreted in the extracellular space as an inactive pro-enzyme named pro-MMP-9 (92 kDa). However, in the extracellular space, other proteinases, like MMP-3 or MMP-2, cleave the inactive form pro-MMP-9 in the active form of 84 kDa. We can see the MMP9 double band only in those plasma samples, which show more significant quantities of MMP9 than the others. However, the analysis of the active band of MMP9 in plasma and its relationship with other metalloproteases may be the subject of future studies.

We added these comments in the discussion of the new version of the manuscript.

3) Please comment on the fact that the patients are not all homozygous for F508-del. Could their second mutation affect the outcome of the pharmacological treatment?

We analysed biological samples of patients submitted to Trikafta therapy selected by our clinicians. The plethora of patients involved in our study displayed different second mutations in the CFTR gene that could affect the outcome of the pharmacological treatment. However, the variability in clinical responses to the therapy of patients with second mutations belonging to the same class led us to find some of the possible molecular mechanisms underlying this variability, and this study was the principal aim of our manuscript. The knowledge obtained with our analyses should be considered to identify the molecular players that could influence the response to Trikafta or other CFTR modulators to accelerate a personalised medicine approach.

The English language is overall very good. There are no major issues but double-check "4.7. Zimography analysis".

As suggested, we reviewed all the manuscript by using a paid editing service Grammarly Premium.

Reviewer 3 Report

The authors investigate the altering effect of elexacaftor/tezacaftor/ivacaftor (ETI) treatment on proinflammatory status of circulating mononuclear cells (CMCs) in patients with CF. They provide evidence, in agreement with the literature, that Matrix metalloprotease 9 (MMP9) is upregulated in CMCs derived from CF compared to healthy individuals. Additionally, Capraro et al. present that intracellular CMC or serum MMP9 levels decline in responders to treatment following 5 or 10 months of ETI administration. Moreover, the manuscript explores NF-kB inactivation by IkBα as a possible mechanism of decline in MMP9 expression by ETI therapy. The authors conclude that ETI stabilizes IkBα with an unknown mechanism of action, which will inactivate NF-kB. Inactivation of NF-kB, a major transcription factor driving MMP9 expression, results in a decline of MMP9 expression.  

Critics:

1)      Throughout the manuscript, a large number of sentences are overexplanatory and long. This makes the message hard to understand that the authors intend to communicate. Please consider rewriting the manuscript. It will highly devaluate the scientific content if the manuscript stays as it is now.

2)      I suggest composing an additional figure describing the proposed pathway involvement in the action of ETI regarding the downregulation of MMP9.

3)      Providing a full dataset regarding an additional non-responder patient with CF is desirable. This would provide a more balanced statistic. The dataset contains only technical repeats from non-responders compared to showing the biological diversity of responders. If possible, please wait for patient 73 to finish the study and include data from this patient in the main study.

4)      Separating the 5- and 10-month treatment samples in Figure 2. B would be desirable. It could show a time course for treatment acting on MMP9 expression.

5)      The actin blot for patient ID CF85 in Figure 1. A shows an almost undetectable actin band regarding lane 3. Could the authors explain the reason behind this? How was this patient sample quantitated (Figure 1. B) and included in the statistics?

6)      It needs to be clarified how WB regarding Figure 3 was conducted. All protein size is approximately in the 40-50KD range, including β-actin. If the loading control was not analyzed from the same blot with the protein of interest, then the quantification could potentially introduce an error not using the same blot. The authors may need to consider a different loading control for this experiment so that the molecular weight does not overlap with the protein of interest.

7)      Genotype listed in Table 1 for patient ID 85 needs to be corrected. It is a typo representing the mutation on the second allele. It is also a typo in the figure legend pmoli /min/103 cells.

 Throughout the manuscript, a large number of sentences are overexplanatory and long. This makes the message hard to understand that the authors intend to communicate. Please consider rewriting the manuscript. It will highly devaluate the scientific content if the manuscript stays as it is now.

Author Response

1)      Throughout the manuscript, a large number of sentences are overexplanatory and long. This makes the message hard to understand that the authors intend to communicate. Please consider rewriting the manuscript. It will highly devaluate the scientific content if the manuscript stays as it is now.

We thank the Reviewer for the suggestion. We reviewed the level of English of the manuscript by using a paid editing service Grammarly Premium. We hope to have reevaluated the scientific content of the manuscript.

2)      I suggest composing an additional figure describing the proposed pathway involvement in the action of ETI regarding the downregulation of MMP9.

As suggested, in the reviewed version of the manuscript we inserted, in Results section, an additional figure (Figure 7) describing the proposed pathway involvement in the action of Trikafta regarding the downregulation of MMP9.

3)      Providing a full dataset regarding an additional non-responder patient with CF is desirable. This would provide a more balanced statistic. The dataset contains only technical repeats from non-responders compared to showing the biological diversity of responders. If possible, please wait for patient 73 to finish the study and include data from this patient in the main study.

We agree with the Reviewer, but all patients included in the study are selected by our clinicians for Trikafta therapy, hypothesising positive outcomes. Unfortunately, the variability in clinical responses to the therapy of the first five patients analysed led us to find the possible molecular mechanisms underlying this variability, and this study was the principal aim of our manuscript. Thus, it is not easy to obtain a more balanced statistic. During our analyses, we found another non-responder patient (patient 73) who did not include in the main study because it needs further investigations. However, we would like to present the data regarding MMP9 activity, even if preliminary, because it strengthens our hypothesis regarding the downregulation of MMP9 as a biomarker of therapy efficacy.

4)      Separating the 5- and 10-month treatment samples in Figure 2. B would be desirable. It could show a time course for treatment acting on MMP9 expression.

As suggested, we separated the 5- and 10-month treatment samples in the new Figure 2B. As reported in the new version of the manuscript there is no time course for treatment acting on MMP9 expression. Indeed, there is no significant difference between POST I and POST II of all samples considered (Fig. 2B).

5)      The actin blot for patient ID CF85 in Figure 1. A shows an almost undetectable actin band regarding lane 3. Could the authors explain the reason behind this? How was this patient sample quantitated (Figure 1. B) and included in the statistics?

We suppose that the Reviewer refers to Fig. 2A. We replaced the actin blot for patient CF 85 and obviously also that one for MMP9. However, all values obtained were included in the statistics.

6)      It needs to be clarified how WB regarding Figure 3 was conducted. All protein size is approximately in the 40-50KD range, including β-actin. If the loading control was not analyzed from the same blot with the protein of interest, then the quantification could potentially introduce an error not using the same blot. The authors may need to consider a different loading control for this experiment so that the molecular weight does not overlap with the protein of interest.

We apologize to the Reviewer because we have omitted to specify that membranes were stripped and re-probed for all primary antibodies used. Now we have inserted a sentence in paragraph 4.6 of the new version of the manuscript. 

7)      Genotype listed in Table 1 for patient ID 85 needs to be corrected. It is a typo representing the mutation on the second allele. It is also a typo in the figure legend pmoli /min/103 cells.

We thank the Reviewer, and we modified the text accordingly.

Comments on the Quality of English Language

 Throughout the manuscript, a large number of sentences are overexplanatory and long. This makes the message hard to understand that the authors intend to communicate. Please consider rewriting the manuscript. It will highly devaluate the scientific content if the manuscript stays as it is now.

We reviewed the level of English of the manuscript by using a paid editing service Grammarly Premium. We hope to have reevaluated the scientific content of the manuscript.

Reviewer 4 Report

The study proposed by Capraro et al presents a high degree of translationality. The data obtained are very interesting and suggest a potential role of MMP-9 as a biomarker of CFTR modulators efficacy, but the authors cannot say "to further validate" (page 9, line 221). For validation of the marker, a valid statistical test is needed, i.e. ROC analysis, on a larger number of cases in order to evaluate the performance of the marker. Therefore some points of the manuscript that are speculative on the basis of the data presented need to be reviewed.

Author Response

The study proposed by Capraro et al presents a high degree of translationality. The data obtained are very interesting and suggest a potential role of MMP-9 as a biomarker of CFTR modulators efficacy, but the authors cannot say "to further validate" (page 9, line 221). For validation of the marker, a valid statistical test is needed, i.e. ROC analysis, on a larger number of cases in order to evaluate the performance of the marker. Therefore some points of the manuscript that are speculative on the basis of the data presented need to be reviewed.

We thank the Reviewer for the helpful and precise comments. Thus, the points of the manuscript that are only speculative for the moment were accordingly reviewed.

Round 2

Reviewer 3 Report

The authors revised the content of the manuscript according to the suggestions. Studies presented by the authors provide importance to the work investigating MMP9 as a novel biomarker for predicting response to modulator therapy.  It is acceptable and understandable to report findings prior to the end of the clinical studies to communicate new findings. Although, the manuscript would be more impactful with the inclusion of additional results.

Minor suggestions:

1)     Please consider breaking up the following sentences into two or more. This approach may serve a better perception by the readers.

·        Lines 50-53: “Thus, to acquire more knowledge about the immune cells and their participation in CF pathophysiology and to determine if and how these cellular phenotypes change in response to CFTR modulator therapy, we have recently analyzed, by a proteomic approach, freshly isolated circulating mononuclear cells (CMCs) following both ex vivo and in vivo ivacaftor treatment.”

·        Lines 57-59: “In line with these results, it has been reported that, based on RNA-Seq evaluation of whole blood gene expression changes in response to lumacaftor/ivacaftor, MMP9 expression was reduced in those patients that positively responded to the therapy.”

·        Lines 66-69: “Finding non-invasive biomarkers of an individual patient's response is very crucial in CF; indeed, although care for CF patients has been revolutionized by the development of CFTR modulators [29,30], clinical responses to CFTR modulator therapy are variable, and some patients do not even derive any benefit from the therapy.

·        Lines 70-74: “Thus, in order to obtain more information about the modulation of MMP9 and confirm that the downregulation of MMP9 in CMCs could be a valuable biomarker of CF modulator therapy efficacy, in this study, we have measured MMP9 levels both in CMCs and plasma of responders and non-responders to Trikafta®, the triple CFTR modulator therapy for the most CF patients.”

2)     Other considerations:

·        Line 56: We should not use the phrase “test patients” with drugs. Instead, we “treat” them. Please consider rephrasing it.

·        Line 249: Please correct the typo NK-kB to NF-kB.

·        Please consider using the phrase "patient with CF" (PwCF) rather than CF patient. The CF community prefers the suggested term.

Please take into consideration breaking up the specified sentences into two or more. This approach may serve a better perception by the readers.

Author Response

Reviewer 3

The authors revised the content of the manuscript according to the suggestions. Studies presented by the authors provide importance to the work investigating MMP9 as a novel biomarker for predicting response to modulator therapy.  It is acceptable and understandable to report findings prior to the end of the clinical studies to communicate new findings. Although, the manuscript would be more impactful with the inclusion of additional results.

Minor suggestions:

1)  Please consider breaking up the following sentences into two or more. This approach may serve a better perception by the readers.

  Lines 50-53: “Thus, to acquire more knowledge about the immune cells and their participation in CF pathophysiology and to determine if and how these cellular phenotypes change in response to CFTR modulator therapy, we have recently analyzed, by a proteomic approach, freshly isolated circulating mononuclear cells (CMCs) following both ex vivo and in vivo ivacaftor treatment.”

We thank the Reviewer for the suggestion. The sentence was rewritten.

Lines 57-59: “In line with these results, it has been reported that, based on RNA-Seq evaluation of whole blood gene expression changes in response to lumacaftor/ivacaftor, MMP9 expression was reduced in those patients that positively responded to the therapy.”

We thank the Reviewer for the suggestion. The sentence was broken up into two.

Lines 66-69: “Finding non-invasive biomarkers of an individual patient's response is very crucial in CF; indeed, although care for CF patients has been revolutionized by the development of CFTR modulators [29,30], clinical responses to CFTR modulator therapy are variable, and some patients do not even derive any benefit from the therapy.

We thank the Reviewer for the suggestion. The sentence was broken up into two.

Lines 70-74: “Thus, in order to obtain more information about the modulation of MMP9 and confirm that the downregulation of MMP9 in CMCs could be a valuable biomarker of CF modulator therapy efficacy, in this study, we have measured MMP9 levels both in CMCs and plasma of responders and non-responders to Trikafta®, the triple CFTR modulator therapy for the most CF patients.”

We thank the Reviewer for the suggestion. The sentence was broken up into two.

2)     Other considerations:

 Line 56: We should not use the phrase “test patients” with drugs. Instead, we “treat” them. Please consider rephrasing it.

We thank the Reviewer for the suggestion. We substituted the term “test” with “treat”.

Line 249: Please correct the typo NK-kB to NF-kB.

We corrected the typo NK-kB to NF-kB

 Please consider using the phrase "patient with CF" (PwCF) rather than CF patient. The CF community prefers the suggested term.

We thank the Reviewer for the suggestion. In the new version of the manuscript we used the phrase “patient with CF” (PwCF) rather than CF patient.

Comments on the Quality of English Language

Please take into consideration breaking up the specified sentences into two or more. This approach may serve a better perception by the readers.